# Factors Associated with Osteoarthritis and Their Influence on Health-Related Quality of Life in Older Adults with Osteoarthritis: A Study Based on the 2020 Korea National Health and Nutrition Examination Survey

**DOI:** 10.3390/ijerph20126073

**Published:** 2023-06-07

**Authors:** Weon-Young Chang, Sungwook Choi, Seung Jin Yoo, Jaeryun Lee, Chaemoon Lim

**Affiliations:** 1Department of Surgery, Jeju National University Hospital, Jeju 63241, Republic of Korea; orkorea@jejunu.ac.kr; 2Department of Orthopedic Surgery, Jeju National University Hospital, Jeju 63241, Republic of Korea; swchoi1115@gmail.com (S.C.); syoo06@gmail.com (S.J.Y.); izz1413@naver.com (J.L.)

**Keywords:** osteoarthritis, health-related quality of life, depression

## Abstract

Numerous studies have investigated factors associated with osteoarthritis (OA), but few have investigated their effects on psychological problems and health-related quality of life in older adults with OA. We aimed to investigate factors associated with OA and their influence on health-related quality of life in older adults with OA. Among 1394 participants aged ≥65 years, 952 and 442 were categorized into the OA and non-OA groups, respectively. Comprehensive data on demographic characteristics, medical conditions, health-related quality of life, blood test results, and nutritional intake were obtained. Univariate and multivariate logistic regression analyses were used to evaluate the odds ratio for factors associated with OA, including age (odds ratio (OR), 1.038; *p* = 0.020), female sex (OR, 5.692; *p* < 0.001), body mass index (OR, 1.108; *p* < 0.001), hypertension (OR, 1.451; *p* < 0.050), hyperlipidemia (OR, 1.725; *p* = 0.001), osteoporosis (OR, 2.451; *p* < 0.001), and depression (OR, 2.358; *p* = 0.041). The OA group showed a significantly lower subjective health status (*p* < 0.001) and higher difficulty in mobility (*p* < 0.001) and pain/discomfort (*p* = 0.010) than the non-OA group. The sleeping hours were significantly shorter in the OA group than those in the non-OA group (*p* = 0.013). OA was a significant contributing factor for unfavorable health-related quality of life in older adults. Controlling the factors associated with OA should be prioritized, and health-related quality of life should be monitored in older adults with OA.

## 1. Introduction

Osteoarthritis (OA) is a common disorder in older adults. It is prevalent in approximately 12.5% of total adults aged >50 years in Korea [1]. The global prevalence of OA ranges from 12.3% to 21.6% [2]. OA is the greatest contributor to medical costs of chronic diseases, accounting for KRW 6.44 million per year [3]. The prevalence of OA increases with age, and the medical costs are expected to rise [4]. Although OA is not a life-threatening disease, it is difficult to treat and requires continuous symptom control. OA is considered a normal aging process; thus, early management and appropriate medical treatment are not performed [5].

OA is a chronic degenerative disease that results in progressive loss of function, pain, and stiffness [6]. Elderly patients with OA can perform decreased levels of physical activities due to persistent joint pain and stiffness and perceive their health condition as worse than those without OA [7,8]. Moreover, elderly patients with OA are more likely to have more depressive symptoms than those without OA because of gait disorders, limited daily activities, and high chance of falling down [9]. The depression in elderly patients with OA results in pain aggravation, adverse effects on postoperative prognosis, and an increase in suicide risk [10].

A number of studies investigated and reported that the factors associated with OA are age, female sex, obesity, hypertension, low educational level, and low income [11,12,13]. However, most of these studies focused on incidence and factors associated with OA. Few studies have investigated the effects of associated factors on psychological problems, including stress and depression, and health-related quality of life in older adults with OA. To reduce the social burden of OA in older adults, psychological problems and health-related quality of life need to be elucidated and the associated demographic factors need to be identified.

Using data from the 2020 Korea National Health and Nutrition Examination Survey (KNHANES), we aimed to investigate factors associated with OA and their influence on health-related quality of life in elderly patients with OA. We anticipated that the findings of this study would enhance the management strategies for OA and ultimately improve the health-related quality of life in older adults with OA.

## 2. Materials and Methods

### 2.1. Data Collection (KNHANES)

This study used data from the 2020 KNHANES, which is a national representative cross-sectional survey managed and released by the Korean Center for Disease Control for Health Statistics. The KNHANES has been performed periodically since 1998 to evaluate the general health status, healthcare utilization and perception, food consumption, and nutritional status of the South Korean population at the national and regional levels. The KNHANES population was selected using proportional allocation systematic sampling with a stratified multistage probability. Standardized interviews were conducted using an established questionnaire to obtain demographic variables, family and medical histories, medications, socioeconomic characteristics (occupation, educational level, household income, and marital status), health-related variables (smoking, alcohol consumption, and exercise), and nutritional variables.

### 2.2. Ethical Consideration

This study was approved by the appropriate Institutional Review Board (Ethics Committee). All methods were performed in accordance with relevant guidelines and regulations (Declaration of Helsinki). The requirement for informed consent was waived by the Institutional Review Board because all participants of KNHANES have already provided written informed consent and all KNHANES data are publicly available.

### 2.3. Study Population

Of the 10,113 potential respondents, 8018 (79.3%) participated in the 2020 KNHANES. We included 1419 participants aged ≥65 years. Of these, 25 did not complete the survey. Thus, 1394 participants were finally included in this study. Based on the questionnaire on the diagnosis of OA, 952 participants were categorized into the OA group, and 442 participants were categorized into the non-OA group.

### 2.4. Covariates

The following baseline demographic characteristics were assessed: age, sex, height, weight, waist circumference, body mass index (BMI), current smoking status, current alcohol consumption, educational level (low: middle school or less; high: high school or more), income level (low: less than 50%; high: 50% or more), and current economic activity.

Medical conditions (hypertension, hyperlipidemia, stroke, myocardial infarction, osteoporosis, diabetes mellitus, gastric cancer, lung cancer, kidney disease, liver cirrhosis, and depression) were identified using a self-report questionnaire based on the diagnoses provided by the doctors.

Blood tests included fasting blood sugar, total cholesterol, high-density lipoprotein cholesterol, low-density lipoprotein cholesterol, triglyceride, aspartate aminotransferase (glutamic oxaloacetic transaminase), alanine transferase (glutamic pyruvic transaminase), hemoglobin, hematocrit, creatinine, blood urea nitrogen, and white blood cell, red blood cell, and platelet counts. Urine tests included urine acidity (pH), specific gravity, protein, glucose, creatinine, sodium, and albumin.

Using a 24 h dietary recall questionnaire, nutritional intake was evaluated based on the nutritional ingredients of the food that the participants had consumed. Nutritional ingredients included total food mass, total calories, total water, protein, fat, carbohydrate, calcium, phosphorus, iron, sodium, potassium, vitamin A, carotene, retinol, vitamin B1 (thiamin), vitamin B2 (riboflavin), vitamin B3 (niacin), and vitamin C.

Health-related quality of life was evaluated using subjective health status, activity limitation, the European Quality of Life-5 Dimensions (EQ-5D), sleeping hours, stress, and the Patient Health Questionnaire-9 (PHQ-9).

EQ-5D is a widely used tool that assesses the quality of life [14]. The EQ-5D consists of following five dimensions: mobility, self-care, usual activity, pain/discomfort, and anxiety/depression. Each dimension has a five-point rating scale: no problems, slight problems, moderate problems, severe problems, and extreme problems. The PHQ-9 is a popular self-reporting tool used to identify major depressive disorders and severity of depressive symptoms [15]. The PHQ-9 has nine questions: lack of interest, depressed mood, sleeping difficulties, tiredness, appetite problems, concentration problems, psychomotor agitation/retardation, negative feelings about self, and suicidal ideation. Each question is assessed on a four-point Likert scale (3 = nearly every day, 2 = more than half of the days, 1 = several days, and 0 = not at all). Out of a total score of 27, the severity of depression according to the score is as follows. Depression severity: 0–4 none, 5–9 mild, 10–14 moderate, 15–19 moderately severe, 20–27 severe. The cutoff value of PHQ-9 was set to 10 [16].

### 2.5. Statistical Analyses

The KNHANES is a statistical survey that was not randomly sampled. This survey was performed using a complex stratified multistage probability sampling model to represent the entire Korean population. The participants were stratified using proportional allocation system sampling and is not equally representative of the Korean population. As a result, each participant had individual power (sample weight). Sample weight was estimated based on the sampling, age, and sex of the reference population. Sample weights were applied to account for the complex sampling methods. Descriptive analysis was performed for all variables, with mean and standard error for continuous data and frequency and percentage for categorical data. Univariate and multivariate logistic regression analyses were used to evaluate the odds ratios (ORs) for factors associated with OA. Multivariate logistic regression analysis was adjusted for age, sex, and BMI. Statistical analyses were performed using IBM SPSS ver. 21.0 for Windows (IBM Corporation, Armonk, NY, USA), with statistical significance set at *p* < 0.05.

## 3. Results

### 3.1. Factors Associated with OA

Among the demographic characteristics, the factors associated with OA were age (adjusted OR, 1.038; 95% confidence interval (CI), 1.006–1.072, *p* = 0.020), female sex (adjusted OR, 5.692; 95% CI, 4.127–7.850, *p* < 0.001), and BMI (adjusted OR, 1.108; 95% CI, 1.056–1.163, *p* < 0.001) (Table 1). Among the medical diseases, the factors associated with OA were hypertension (adjusted OR, 1.451; 95% CI, 1.080–1.949, *p* = 0.050), hyperlipidemia (adjusted OR, 1.725; 95% CI, 1.270–2.344, *p* = 0.001), osteoporosis (adjusted OR, 2.451; 95% CI, 1.684–3.569, *p* < 0.001), and depression (adjusted OR, 2.358; 95% CI, 1.037–5.362, *p* = 0.041) (Table 2). There was no significant difference in blood test and nutritional intake factors between OA group and non-OA group (Table 3 and Table 4).

### 3.2. Health-Related Quality of Life

OA adversely affects health-related quality of life. The OA group had a significantly lower subjective health status than the non-OA group (*p* < 0.001). Moreover, the OA group had significantly more activity limitation than the non-OA group (*p* < 0.001). According to the EQ-5D results, the OA group had more pain/discomfort (*p* = 0.010) and anxiety/depression (*p* = 0.037) than the non-OA group. The sleeping hours were significantly shorter in the OA group than those in the non-OA group (*p* = 0.013). The OA group had significantly more stress than the non-OA group (*p* = 0.025) (Table 5).

## 4. Discussion

Using the data based on 2020 KNHANES, this study evaluated the factors associated with OA and their influence on health-related quality of life in older adults with OA. Older age, female sex, and high BMI were significant factors associated with OA. Hypertension, hyperlipidemia, osteoporosis, and depression were also significantly associated with OA. Moreover, OA adversely affected health-related quality of life in older adults.

In this study, we confirmed that demographic characteristics of age, female sex, and BMI were factors associated with OA. Age and female are well-known factors associated with OA in previous studies [17,18]. BMI is another significant factor associated with OA [19]. High BMI and obesity can cause OA by increasing the mechanical load on weight-bearing joints [20]. Although obesity does not necessarily cause mechanical loading on joints in the hand, obesity is known to be associated with hand OA [21,22]. Obesity contributes to development of OA by increasing mechanical loading on joints during weight bearing. Moreover, metabolic factors, including elevated adipocytokine levels, mediate proinflammatory response associated with OA [23]. Educational level and income were not identified as risk factors for OA in this study, although a few previous studies reported an association between low socioeconomic status and OA [24,25,26]. Callahan et al. reported that low educational level is significantly associated with OA [25]. A low educational level could cause reduced health-promoting activities and health literacy. This difference in lifestyle could be responsible for the different incidences of OA [27]. However, studies showed inconsistent results in socioeconomic status, and no consensus has been reached on low socioeconomic status. Further studies on the effect of socioeconomic status on OA are required.

This study demonstrated that metabolic diseases, such as hypertension and hyperlipidemia, are associated with OA. Several studies also described an association between metabolic diseases and OA. This association could be related to inflammation, lipid metabolism, and cytokines [28,29]. Metabolic diseases and OA are known to share the same mechanism of inflammation, which could lead to a chronic low-grade inflammatory response in the joints [30,31]. Moreover, in hyperlipidemia, cholesterol could accumulate in joint cartilage and impair the efflux function of the cartilage, which eventually leads to OA [32]. Several studies reported that proinflammatory cytokines, such as interleukin-6, play an important role in hypertension and OA [33,34]. However, further studies are warranted to understand these mechanisms.

Osteoporosis was identified as another factor associated with OA in this study. Although osteoporosis and OA are associated with aging processes, bone, and cartilage, they are completely different diseases [35,36]. Moreover, a negative relationship between osteoporosis and OA has been widely accepted, as an increased physical load is speculated to be a protective factor for osteoporosis and a negative factor for OA [37,38,39]. However, age and female sex are common risk factors for both osteoporosis and OA [18,40]. In this study, multivariate logistic regression analysis showed that osteoporosis was significantly associated with OA when adjusted for age, sex, and BMI. A recent study has shown that OA may have a positive protective effect on osteoporosis fracture when patients with OA suffer osteoporosis fracture [41]. However, while osteoporosis medication reduces arthritis pain, it does not prevent the structural progression of hip arthritis [42]. As a result, the relationship between osteoporosis and OA remains controversial and requires further investigation.

This study demonstrated that depression is a factor associated with OA. Depression is one of the most common comorbidities with OA and affects OA symptom progression and quality of life [43]. Agustini et al. found an OR of 1.41 (CI, 1.27–1.57) for depression in patients with OA compared to that in patients without OA [44]. Wang et al. also found that patients with depressive symptoms have two or three times higher risk for developing OA than those without depression [45]. Moreover, depression may lead to increased OA pain and can be a predictor of progression of OA pain [46]. Recently, several studies have suggested that depression increases the health-related burden of OA and aggravates cartilage degeneration due to an increased systemic inflammation response. These findings suggest that depression is associated with symptom progression and structural progression of OA [43]. However, some studies found no significant association between symptom progression or structural progression of OA and depression [47]. Thus, the relationship between depression and OA remains inconclusive.

In this study, OA adversely affected health-related quality of life in older adults. Patients with OA showed significantly lower subjective health status, higher difficulty in activity and pain/discomfort, and fewer sleeping hours than patients without OA. Previous studies showed an association between OA and health-related quality of life, which supports our findings. Vernoses et al. reported that patients with OA tend to have more depressive symptoms than those without OA. They also found that health-related quality of life, including pain and functional limitation, is strongly associated with depressive symptoms [48]. Sugai et al. showed that elderly patients with severe pain and dysfunction associated with OA are at a significantly increased risk of having depressive symptoms. They found that going outdoors becomes a burden for elderly patients with OA, and they are likely to have depression [49]. Therefore, increased attention should be focused on health-related quality of life and psychiatric outcomes in elderly patients with OA.

This study has some limitations. First, the KNHANES was conducted using self-reported medical conditions. The validity of the self-reported measures used in this study was not verified. Second, treatments, including medication, operation, and alternative therapy for OA, were not included in this study, which could affect health-related quality of life. Third, although many covariates were adjusted in this study, several residual or hidden confounding covariates were not excluded. Finally, OA in this study was not classified according to body part. Moreover, the duration and severity of OA were not included in this study. Further studies on OA in specific body parts, duration, and severity are required.

## 5. Conclusions

We confirmed that older age, female sex, and BMI are significant factors associated with OA. Moreover, hypertension, hyperlipidemia, osteoporosis, and depression are significantly associated with OA. Medical treatments for these associated factors may be important to manage OA. OA is a significant contributing factor to an unfavorable health-related quality of life in older adults, including low subjective health status, difficulty in mobility, and decreased sleeping hours. Increased attention should be focused on controlling factors associated with OA, and health-related quality of life should be monitored in elderly patients with OA.

## Figures and Tables

**Table 1 ijerph-20-06073-t001:** Comparison of demographic characteristics between non-OA group and OA group.

	Total (*n* = 1394) (Weighted *n* = 7,015,592)	Non-OA Group (*n* = 952) (Weighted *n* = 4,676,648)	OA Group (*n* = 442) (Weighted *n* = 2,338,944)	Univariate *p*	Multivariate *p*	OR (95% CI)
Age (years)	72.5 ± 0.2	72.2 ± 0.3	73.0 ± 0.3	0.065	0.020	1.038 (1.006–1.072)
Sex						
- Female (%)	53.9	41.1	79.6	<0.001	<0.001	5.692 (4.127–7.850)
- Male (%)	46.1	58.9	20.4			
Height (cm)	159.1 ± 0.3	160.8 ± 0.3	155.7 ± 0.4	<0.001	0.483	1.010 (0.983–1.037)
Weight (kg)	61.1 ± 0.4	61.6 ± 0.5	60.2 ± 0.4	0.030	<0.001	1.039 (1.024–1.054)
Waist circumference	87.4 ± 0.4	87.0 ± 0.5	88.2 ± 0.4	0.035	<0.001	1.040 (1.024–1.056)
BMI (kg/m^2^)	24.1 ± 0.1	23.8 ± 0.2	24.7 ± 0.2	<0.001	<0.001	1.108 (1.056–1.163)
Current smoker (%)	10.1	13.2	3.9	<0.001	0.093	0.525 (0.247–1.116)
Current alcohol use (%)	45.8	50.0	37.4	0.001	0.687	0.936 (0.677–1.294)
Education level						
- Low (%)	67.9	62.2	79.3	<0.001	0.075	0.830 (0.677–1.019)
- High (%)	32.1	37.8	20.7			
Income						
- Low (%)	48.5	47.2	49.6	0.493	0.352	0.935 (0.812–1.078)
- High (%)	51.5	52.8	50.4			
Economic activity						
- Yes (%)	36.1	37.4	33.3	0.212	0.075	0.830 (0.677–1.019)
- No (%)	63.9	62.6	66.7			

Continuous values are presented as mean and standard error (mean ± SE); categorical parameters are presented as count with percentage (%), OR: odds ratio, CI: confidence interval, BMI: body mass index.

**Table 2 ijerph-20-06073-t002:** Comparison of medical diseases between non-OA group and OA group.

	Total (*n* = 1394) (Weighted *n* = 7,015,592)	Non-OA Group (*n* = 952) (Weighted *n* = 4,676,648)	OA Group (*n* = 442) (Weighted *n* = 2,338,944)	Univariate *p*	Multivariate *p*	OR (95% CI)
HTN (%)	51.3	47.3	59.3	0.003	0.014	1.451 (1.080–1.949)
Hyperlipidemia (%)	33.5	28.3	44.0	<0.001	0.001	1.725 (1.270–2.344)
Stroke (%)	1.6	1.8	1.2	0.520	0.823	1.149 (0.339–3.892)
Myocardial infarction (%)	3.9	4.2	3.1	0.520	0.979	0.988 (0.419–2.329)
Osteoporosis (%)	17.3	9.7	32.6	<0.001	<0.001	2.451 (1.684–3.569)
Diabetes mellitus (%)	22.2	21.5	23.7	0.823	0.741	1.042 (0.815–1.331)
Gastric cancer (%)	0.4	0.5	0.2	0.463	0.693	0.692 (0.109–4.370)
Lung cancer (%)	0.1	0.1	0.1	0.318	0.176	4.487 (0.505–39.890)
Kidney disease (%)	1.9	1.9	2.0	0.882	0.655	1.245 (0.473–3.283)
Liver cirrhosis (%)	0.2	0.1	0.3	0.371	0.714	1.534 (0.156–15.122)
Depression (%)	3.2	2.0	5.7	0.006	0.041	2.358 (1.037–5.362)

Categorical parameters are presented as count with percentage (%), OR: odds ratio, CI: confidence interval, HTN: hypertension.

**Table 3 ijerph-20-06073-t003:** Comparison of laboratory tests between non-OA group and OA group.

	Total (*n* = 1394) (Weighted *n* = 7,015,592)	Non-OA Group (*n* = 952) (Weighted *n* = 4,676,648)	OA Group (*n* = 442) (Weighted *n* = 2,338,944)	Univariate *p*	Multivariate *p*	OR (95% CI)
Blood test						
FBS (mg/dL)	106.2 ± 0.8	107.4 ± 1.0	103.7 ± 1.0	0.019	0.059	1.007 (1.000–1.014)
Total cholesterol (mg/dL)	178.7 ± 1.4	179.2 ± 1.7	177.6 ± 2.3	0.531	0.052	1.004 (1.000–1.007)
HDL cholesterol (mg/dL)	48.9 ± 0.5	48.2 ± 0.7	50.4 ± 0.7	0.032	0.558	0.995 (0.979–1.012)
LDL cholesterol (mg/dL)	103.9 ± 3.0	103.7 ± 3.7	104.4 ± 4.2	0.904	0.334	1.006 (0.994–1.019)
Triglycerides (mg/dL)	124.0 ± 2.6	124.7 ± 3.5	122.6 ± 3.6	0.695	0.925	1.000 (0.997–1.002)
AST (SGOT, IU/L)	25.8 ± 0.3	26.1 ± 0.4	25.2 ± 0.5	0.218	0.502	1.006 (0.989–1.023)
ALT (SGPT, IU/L)	21.1 ± 0.4	21.6 ± 0.5	20.0 ± 0.6	0.056	0.826	0.999 (0.986–1.012)
Hemoglobin (g/dL)	13.4 ± 0.1	13.6 ± 0.1	12.9 ± 0.1	<0.001	0.135	1.074 (0.978–1.180)
Hematocrit (%)	40.8 ± 0.2	41.4 ± 0.3	39.6 ± 0.2	<0.001	0.363	1.015 (0.983–1.048)
Creatinine	0.9 ± 0.0	0.9 ± 0.0	0.8 ± 0.0	0.180	0.228	0.781 (0.521–1.170)
BUN (mg/dL)	18.1 ± 0.2	17.9 ± 0.3	18.3 ± 0.3	0.320	0.079	1.096 (0.990–1.215)
WBC count (thousands/uL)	6.2 ± 0.1	6.2 ± 0.1	6.0 ± 0.1	0.080	0.776	1.011 (0.935–1.095)
RBC count (millions/uL)	4.3 ± 0.0	4.4 ± 0.0	4.2 ± 0.0	<0.001	0.192	1.208 (0.908–1.607)
PLT	234.8 ± 2.3	232.4 ± 2.5	239.8 ± 3.9	0.093	0.481	1.001 (0.998–1.003)
Urine test						
pH	6.0 ± 0.0	6.0 ± 0.0	6.0 ± 0.1	0.219	0.780	0.976 (0.820–1.162)
Urine specific gravity	1.0 ± 0.0	1.0 ± 0.0	1.0 ± 0.0	0.999	0.999	0.000
Urine protein	0.2 ± 0.0	0.2 ± 0.0	0.2 ± 0.0	0.037	0.422	1.100 (0.870–1.391)
Urine glucose	0.3 ± 0.0	0.4 ± 0.0	0.2 ± 0.0	0.032	0.102	1.164 (0.970–1.397)
Urine creatinine	90.5 ± 1.8	94.7 ± 2.3	82.1 ± 2.8	0.003	0.837	1.000 (0.997–1.004)
Urine sodium	115.1 ± 1.5	115.2 ± 1.8	114.9 ± 2.4	0.910	0.163	0.998 (0.994–1.001)
Urine albumin	36.2 ± 6.4	39.5 ± 8.9	29.9 ± 7.6	0.428	0.957	1.000 (0.999–1.001)

Continuous values are presented as mean and standard error (mean ± SE); OR: odds ratio, CI: confidence interval, FBS: fasting blood sugar, HDL: high density lipoprotein, LDL: low density lipoprotein, AST: aspartate aminotransferase, ALT: alanine aminotransferase, BUN: blood urea nitrogen, WBC: white blood cell, RBC: red blood cell, PLT: platelet, pH: hydrogen exponent.

**Table 4 ijerph-20-06073-t004:** Comparison of nutrition intake between non-OA group and OA group.

	Total (*n* = 1394) (Weighted *n* = 7,015,592)	Non-OA Group (*n* = 952) (Weighted *n* = 4,676,648)	OA Group (*n* = 442) (Weighted *n* = 2,338,944)	Univariate *p*	Multivariate *p*	OR (95% CI)
Total food mass (g)	1252.6 ± 33.2	1288.6 ± 35.4	1180.6 ± 42.8	0.014	0.677	1.000 (1.000–1.000)
Total calories (kcal)	1555.5 ± 28.1	1592.0 ± 29.6	1482.5 ± 43.5	0.028	0.079	1.000 (1.000–1.000)
Total water (g)	844.4 ± 26.7	868.3 ± 29.1	796.5 ± 33.1	0.034	0.927	1.000 (1.000–1.000)
Protein (g)	54.5 ± 1.2	55.6 ± 1.3	52.4 ± 2.2	0.212	0.083	0.995 (0.989–1.001)
Fat (g)	29.7 ± 1.1	30.6 ± 1.2	28.01 ± 1.4	0.125	0.536	0.998 (0.991–1.005)
Carbohydrates (g)	258.9 ± 4.1	262.8 ± 4.4	251.0 ± 6.4	0.105	0.144	0.998 (0.997–1.000)
Calcium (mg)	450.1 ± 12.4	454.9 ± 13.7	442.3 ± 18.3	0.526	0.142	1.000 (0.999–1.000)
Phosphorus (mg)	886.1 ± 18.5	905.6 ± 20.2	847.0 ± 29.3	0.086	0.235	1.000 (0.999–1.000)
Iron (mg)	10.2 ± 0.3	10.4 ± 0.3	9.8 ± 0.4	0.266	0.190	0.985 (0.963–1.008)
Sodium (mg)	2836.2 ± 71.9	2943.6 ± 76.8	2621.5 ± 128.2	0.042	0.240	1.000 (1.000–1.000)
Potassium (mg)	2663.3 ± 54.6	2738.3 ± 64.7	2513.3 ± 77.6	0.020	0.531	1.000 (1.000–1.000)
Vit A (μg)	316.8 ± 11.4	326.8 ± 14.6	297.0 ± 16.7	0.210	0.754	1.000 (0.999–1.001)
Carotene (μg)	2661.4 ± 110.9	2748.3 ± 142.9	2487.5 ± 179.4	0.321	0.744	1.000 (1.000–1.000)
Retinol (μg)	95.1 ± 7.0	97.7 ± 8.8	89.7 ± 7.4	0.382	0.974	1.000 (0.999–1.001)
Vit B1 (thiamine, mg)	1.1 ± 0.0	1.1 ± 0.0	1.0 ± 0.0	0.008	0.579	1.104 (0.777–1.569)
Vit B2 (riboflavin, mg)	1.3 ± 0.0	1.3 ± 0.0	1.2 ± 0.1	0.165	0.346	0.887 (0.690–1.140)
Vit B3 (niacin, mg)	9.8 ± 0.2	10.2 ± 0.3	9.1 ± 0.3	0.004	0.947	0.999 (0.970–1.029)
Vit C (mg)	60.3 ± 2.6	62.9 ± 3.4	55.0 ± 3.7	0.188	0.244	1.002 (0.999–1.005)

Continuous values are presented as mean and standard error (mean ± SE), OR: odds ratio, CI: confidence interval, Vit: vitamin.

**Table 5 ijerph-20-06073-t005:** Comparison of health-related quality of life between non-OA group and OA group.

	Total (*n* = 1394) (Weighted *n* = 7,015,592)	Non-OA Group (*n* = 952) (Weighted *n* = 4,676,648)	OA Group (*n* = 442) (Weighted *n* = 2,338,944)	Univariate *p*	Multivariate *p*	OR (95% CI)
Subjective health status (very poor/poor/fair/good/very good)	3.1 ± 0.0	2.9 ± 0.0	3.3 ± 0.1	<0.001	<0.001	1.411 (1.207–1.650)
Activity limitation (%)	14.4	10.9	21.3	<0.001	<0.001	2.088 (1.408–3.096)
EuroQoL						
- Mobility	1.4 ± 0.03	1.3 ± 0.03	1.6 ± 0.04	0.019	0.063	0.753 (0.558–1.016)
- Self care	1.1 ± 0.02	1.1 ± 0.02	1.2 ± 0.04	0.069	0.256	0.879 (0.702–1.099)
- Usual activities	1.2 ± 0.03	1.2 ± 0.03	1.3 ± 0.04	0.041	0.163	0.841 (0.659–1.074)
- Pain/discomfort	1.4 ± 0.02	1.2 ± 0.02	1.6 ± 0.05	0.001	0.010	1.625 (1.125–2.346)
- Anxiety/depression	1.2 ± 0.02	1.2 ± 0.02	1.3 ± 0.05	0.038	0.037	1.106 (1.001–1.308)
Sleeping hours	6.5 ± 0.1	6.7 ± 0.1	6.2 ± 0.1	0.008	0.013	0.846 (0.757–0.945)
Stress (%)	15.5	12.2	22.2	<0.001	0.025	1.665 (0.067–2.599)
PHQ-9 PHQ-9 > 10 (%)	2.1 ± 0.1 4.9	1.8 ± 0.1 3.8	2.8 ± 0.3 7.2	0.001 0.007	0.019 0.018	1.051 (1.008–1.096) 2.202 (1.151–4.215)

Continuous values are presented as mean and standard error (mean ± SE); categorical parameters are presented as count with percentage (%), OR: odds ratio, CI: confidence interval, PHQ-9: patient health questionnaire-9.

## Data Availability

The data presented in this study are openly available at http://www.kdca.ro.kr (accessed on 10 April 2023).

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
