# Peer review of "Factors Associated with Osteoarthritis and Their Influence on Health-Related Quality of Life in Older Adults with Osteoarthritis: A Study Based on the 2020 Korea National Health and Nutrition Examination Survey"

_ijerph, 2023, doi:10.3390/ijerph20126073_

Round 1

Reviewer 1 Report

Thank you for the invitation to review this article that aimed to identify risk factors associated with OA and their association with HRQOL in a population-based sample. I enjoyed reading this manuscript, although there are several points for clarification.

1.       The title should be changed as this is a cross sectional study and therefore not possible to determine causality. Rather than risk factors, association would be a more appropriate word. For example, weight although was significantly different, it is lower in the OA group, suggesting that the OA group may have tried weight loss? Furthermore, are the date of diagnosis for the comorbid conditions known? If they occurred after OA, it is not a risk factor. Therefore, the use of the word ‘risk’ should be also carefully considered in the manuscript.

2.       Back to the issue of weight, it is interesting to note in the OA group that the mean weight is lower but have a higher mean BMI. What could be the reason for this difference? Is it possible to also show the frequency by BMI categories?

3.       How is the PHQ-9 score interpreted? Does a lower score indicate fewer depressive symptoms? Is there a cut-off for the PhQ-9? What about using the categorical variable in the model?

4.       The OA group is more likely to be female and gender has been associated with HRQOL. Just out of curiosity, would it be an idea to do subgroup analyses stratified by gender on the association between OA and HRQOL outcomes? The authors discussed that education was not associated with OA after adjustment, contrary to previous studies. What is the frequency of education according to gender? Would females be more likely in the lower educated group? Taken together, such information may be helpful in planning personalized supportive care.

Author Response

Dear Reviewer,

First of all, we are greatly appreciated for the meticulous review and for the opportunity for the revision. We corrected and revised our manuscript based on your suggestions and recommendations. We attempted to revise as recommended and marked the revised section with tracking in the manuscript. Descriptions on revision regarding the reviewer’s comments and suggestions are detailed below. We are greatly appreciated for working with the prestigious journal of yours and will be looking forward to your final decision.

Response to review

  1. The title should be changed as this is a cross sectional study and therefore not possible to determine causality. Rather than risk factors, association would be a more appropriate word. For example, weight although was significantly different, it is lower in the OA group, suggesting that the OA group may have tried weight loss? Furthermore, are the date of diagnosis for the comorbid conditions known? If they occurred after OA, it is not a risk factor. Therefore, the use of the word ‘risk’ should be also carefully considered in the manuscript

-> We appreciated and agreed with your comment. We changed the ‘risk factor’ to ‘association factor’ both in the tile and in the manuscript.

  1. Back to the issue of weight, it is interesting to note in the OA group that the mean weight is lower but have a higher mean BMI. What could be the reason for this difference? Is it possible to also show the frequency by BMI categories?

-> We appreciated your valuable comment. The proportion of female in OA group is significantly higher than non-OA group (79.6%, 41.1% respectively). In general, the weight of female is lower than male, so the mean weight of OA group was lower than non-OA group. On the other hand, in general, the height of female was smaller than male, so the mean height of OA group was smaller than non-OA group. However, the mean BMI of OA group was significantly higher than non-OA group. So, we though the BMI is an association factor for OA.

  1. How is the PHQ-9 score interpreted? Does a lower score indicate fewer depressive symptoms? Is there a cut-off for the PhQ-9? What about using the categorical variable in the model?

-> We appreciated your valuable question. The PHQ-9 is a multipurpose instrument for screening, diagnosing, monitoring and measuring the severity of depression. It is consisted of nine question which scores the depressive symptom as “0” (not at all) to “3” (nearly every day). Out of a total score of 27, the severity of depression according to the score is as follows. Depression severity: 0-4 none, 5-9 mild, 10-14 moderate, 15-19 moderately severe, 20-27 severe. In this study, the mean score was low because the most of the subject had no depressive symptom. The cutoff value of PHQ-9 was set to 10 according to the meta-analysis paper. (line 118-120) Then, we performed logistic regression analysis with the categorical variable of PHQ-9. The OA group had a significantly more depressive symptom than the non-OA group (p = 0.018). (Table 5) (Please see attached eord file.)

  1. The OA group is more likely to be female and gender has been associated with HRQOL. Just out of curiosity, would it be an idea to do subgroup analyses stratified by gender on the association between OA and HRQOL outcomes? The authors discussed that education was not associated with OA after adjustment, contrary to previous studies. What is the frequency of education according to gender? Would females be more likely in the lower educated group? Taken together, such information may be helpful in planning personalized supportive care.

-> We appreciated your valuable comment. We performed subgroup analysis stratified by gender on the association between OA and HRQOL. We added this result as a supplementary data in this paper. (Please see attached eord file.)

We performed subgroup analysis stratified by gender on the association between OA and education. Females are more in the lower education level. However, there was no significant difference between non-OA group and OA group in female gender. (Please see attached eord file.)

Reviewer 2 Report

The manuscript reported the findings of the risk factors for osteoarthritis (OA) and their effects on the health-related quality of life in older patients with OA based on the data from the 2020 Korea National Health and Nutrition Examination Survey. The following are my specific comments for the manuscript:

1. Please use non-biased language in the title and throughout the manuscript, and replace the term 'elderly' with 'older adults' when referring to the study population. This will ensure that the language used is inclusive, respectful, and avoids any negative connotations associated with age.

2. Abstract: Please provide some of the numerical results, such as the odds ratio and p-value, in the abstract to make the abstract more informative.

3. Introduction (Line 52): It seems that the word 'hypothesized' is used incorrectly in the manuscript. A hypothesis is a testable prediction that can be supported or refuted by data. However, in this study, the authors did not test the hypothesis that the results would improve the management of OA to enhance health-related quality of life in older adults with OA. Instead, the study aimed to investigate the risk factors for OA and their effects on health-related quality of life in older adults with OA.

4. Methods: Please provide references for the EQ-5D and PHQ-9 after their description.

5. Methods: The citation information for SPSS is incorrect. It should be “Statistical analyses were performed using SPSS version 21.0 for Windows (IBM Corporation, Armonk, NY, USA).”

6. Methods: What is the rationale of not using stepwise procedure in the multiple logistic regression analysis?

7. Methods: Please explain why standard error instead of standard deviation is used for the description of continuous data? In general, when presenting descriptive statistics such as mean, median, and standard deviation for a sample, it is appropriate to use standard deviation in the table of results.

8. Table 1: The categorical variable “gender” (should be “sex”) should not have an associated standard deviation.

9. Results and tables: Regarding Tables 2 to 5, it's unclear whether the significant demographic variables (age, sex, and BMI) were included in the multiple regression model. Please clarify whether these variables were included as covariates or if the regression models were adjusted for these variables. In fact, the authors can consider using a hierarchical regression model to build their model. In this approach, demographic variables would be entered as covariates in the first block of the regression, followed by other predictors in subsequent blocks, including diseases, lab test results, nutrition intake, and quality of life. This would allow the authors to examine the unique contribution of each variable to the outcomes of interest while controlling for the effects of demographic variables.

10. Conclusion (Line 239): It is not clear why the authors suggested that low protein intake was significantly associated with OA. This result was not mentioned in the Results section. The adjusted odds ratio for protein intake was associated with a p-value of 0.083.

11. The section on limitations (line 245-252): This paragraph should be moved to the end of the Discussion section. The authors may also want to mention that the study did not collect information on the duration or severity of OA, which could affect the health-related quality of life of older adults.

12. References: Please correct all the journal titles that are incorrectly started with a “J”.

Author Response

Dear Reviewer,

First of all, we are greatly appreciated for the meticulous review and for the opportunity for the revision. We corrected and revised our manuscript based on your suggestions and recommendations. We attempted to revise as recommended and marked the revised section with tracking in the manuscript. Descriptions on revision regarding the reviewer’s comments and suggestions are detailed below. We are greatly appreciated for working with the prestigious journal of yours and will be looking forward to your final decision.

Response to review

Reviewer2

  1. Please use non-biased language in the title and throughout the manuscript, and replace the term 'elderly' with 'older adults' when referring to the study population. This will ensure that the language used is inclusive, respectful, and avoids any negative connotations associated with age.

 -> We appreciated and agreed with your comment. We changed the ‘elderly’ to ‘older adults’ both in the tile and in the manuscript.

  1. Abstract: Please provide some of the numerical results, such as the odds ratio and p-value, in the abstract to make the abstract more informative.

 -> We appreciated your comment. We added the odds ratio and p-value in the abstract.

  1. Introduction (Line 52): It seems that the word 'hypothesized' is used incorrectly in the manuscript. A hypothesis is a testable prediction that can be supported or refuted by data. However, in this study, the authors did not test the hypothesis that the results would improve the management of OA to enhance health-related quality of life in older adults with OA. Instead, the study aimed to investigate the risk factors for OA and their effects on health-related quality of life in older adults with OA.

  -> We appreciated and agreed with your comment. We changed the ‘hypothesized’ to ‘thought’. (line 57)

  1. Methods: Please provide references for the EQ-5D and PHQ-9 after their description.

 -> We appreciated your comment. We added the reference of EQ-5D and PHQ-9 as reference number 14 and 15 respectively. (line 109, 114)

  1. Methods: The citation information for SPSS is incorrect. It should be “Statistical analyses were performed using SPSS version 21.0 for Windows (IBM Corporation, Armonk, NY, USA).”

-> We appreciated your comment. We corrected the sentence according to your comment. (line 133)

  1. Methods: What is the rationale of not using stepwise procedure in the multiple logistic regression analysis?

-> We appreciated your valuable comment. In this study, we used multiple logistic regression to obtain the odds ratio. We used the backward elimination method using the odds ratio. In the first step, other predictors (diseases, lab test results, nutrition intake, and quality of life) were put  with demographic variables (age, sex, and BMI) and a model was obtained. Then, we stopped at step 3 by removing unlikely variables through a odds ratio.

  1. Methods: Please explain why standard error instead of standard deviation is used for the description of continuous data? In general, when presenting descriptive statistics such as mean, median, and standard deviation for a sample, it is appropriate to use standard deviation in the table of results.

-> In analysis of complex sample (complex stratified multistage probability sampling model), standard error is used for the description of continuous data.

  1. Table 1: The categorical variable “gender” (should be “sex”) should not have an associated standard deviation.

-> We appreciated and agreed with your comment. We deleted the standard deviation in sex. (Table 1)

  1. Results and tables: Regarding Tables 2 to 5, it's unclear whether the significant demographic variables (age, sex, and BMI) were included in the multiple regression model. Please clarify whether these variables were included as covariates or if the regression models were adjusted for these variables. In fact, the authors can consider using a hierarchical regression model to build their model. In this approach, demographic variables would be entered as covariates in the first block of the regression, followed by other predictors in subsequent blocks, including diseases, lab test results, nutrition intake, and quality of life. This would allow the authors to examine the unique contribution of each variable to the outcomes of interest while controlling for the effects of demographic variables.

 -> We appreciated your valuable comment. In this study, we used multiple logistic regression to obtain the odds ratio. We used the backward elimination method using the likelihood ratio. In the first step, other predictors (diseases, lab test results, nutrition intake, and quality of life).  were put in with demographic variables (age, sex, and BMI) and a model was obtained. Then, we stopped at step 3 by removing unlikely variables through a odds ratio.

  1. Conclusion (Line 239): It is not clear why the authors suggested that low protein intake was significantly associated with OA. This result was not mentioned in the Results section. The adjusted odds ratio for protein intake was associated with a p-value of 0.083.

-> We appreciated and agreed with your comment. We deleted the low protein intake in the conclusion. The adjusted odds ratio for protein intake was statistically significant.

  1. The section on limitations (line 245-252): This paragraph should be moved to the end of the Discussion section. The authors may also want to mention that the study did not collect information on the duration or severity of OA, which could affect the health-related quality of life of older adults.

 -> We appreciated your valuable comment. We moved the limitation to the end of the discussion section. And we mentioned that this study did not collect information on the duration or severity of OA. (line 246-254)

  1. References: Please correct all the journal titles that are incorrectly started with a “J”

 -> We appreciated your valuable comment. We corrected the name of reference journal according to your comment.

Reviewer 3 Report

Risk Factors for Osteoarthritis and their Influence on Health- 2 related Quality of Life in Elderly Patients with Osteoarthritis: 3 A Study Based on the 2020 Korea National Health and Nutri- 4 tion Examination Survey

This article presents the possible effects of osteoarthritis on the quality of life of elderly patients from Korea, including information regarding the risk factors for osteoarthritis among these patients. The article under review demonstrates a high level of proficiency in the use of the English language. The authors have done an excellent job in presenting their research findings in a clear and concise manner, making the article easy to read and understand.

 I recommend taking into consideration the following regards:

  • r28: I recommend providing the worldwide osteoarthritis prevalence.
  • You should mention if you can, in which possible way the obtained results can improve the management of osteoarthritis - if it means involves adding medication or non-pharmacological treatment.  

Author Response

Dear Reviewer,

First of all, we are greatly appreciated for the meticulous review and for the opportunity for the revision. We corrected and revised our manuscript based on your suggestions and recommendations. We attempted to revise as recommended and marked the revised section with tracking in the manuscript. Descriptions on revision regarding the reviewer’s comments and suggestions are detailed below. We are greatly appreciated for working with the prestigious journal of yours and will be looking forward to your final decision.

Response to review

Reviewer3

  1. r28: I recommend providing the worldwide osteoarthritis prevalence.

-> We appreciated your valuable comment. We added the global incidence of OA. (line 32-33)

  1. You should mention if you can, in which possible way the obtained results can improve the management of osteoarthritis - if it means involves adding medication or non-pharmacological treatment.  

-> We appreciated your valuable comment. In conclusion, we added the comment about medical treatment for associated factors (hypertension, hyperlipidemia, osteoporosis, depression). (line 259-260)

Round 2

Reviewer 2 Report

The authors have adequately addressed and incorporated most of my comments and suggestions, except that the response regarding the specific details of the logistic regression, particularly, the utilization of backward elimination method using the likelihood ratio, has not been incorporated in the statistical analyses section of the revised manuscript.

1. The revised title "Association Factors for Osteoarthritis and their Influence on Health-related Quality of Life in Older Adults with Osteoarthritis: A Study Based on the 2020 Korea National Health and Nutrition Examination Survey" can be further improved.

For example, "Factors Associated with Osteoarthritis and their Influence on Health-related Quality of Life in Older Adults with Osteoarthritis: A Study Based on the 2020 Korea National Health and Nutrition Examination Survey" should be easier to understand. Please revise all instances of the word "association" in the abstract and in the main text.

2.  Line 57: " We thought that the results... " should be changed to " We hoped that the results...". Or "We anticipated that the findings of this study would enhance management strategies for osteoarthritis (OA) and ultimately improve the health-related quality of life in older adults affected by OA."

3. Line 133: "SPSS ver. 21.0 for Windows" should be "IBM SPSS ver. 21.0 for Windows"

Author Response

Dear Reviewer,

We are greatly appreciated for the meticulous review and for the opportunity for the minor revision. We corrected and revised the manuscript based on your suggestions and recommendations. We attempted to revise as recommended and marked the revised section with tracking in the manuscript. Descriptions on revision regarding the reviewer’s comments and suggestions are detailed below. We are greatly appreciated for working with the prestigious journal of yours and will be looking forward to your final decision.

Response to review

Reviewer 2

  1. The revised title "Association Factors for Osteoarthritis and their Influence on Health-related Quality of Life in Older Adults with Osteoarthritis: A Study Based on the 2020 Korea National Health and Nutrition Examination Survey" can be further improved. For example, "Factors Associated with Osteoarthritis and their Influence on Health-related Quality of Life in Older Adults with Osteoarthritis: A Study Based on the 2020 Korea National Health and Nutrition Examination Survey" should be easier to understand. Please revise all instances of the word "association" in the abstract and in the main text.

 -> We appreciated your comment. We changed the ‘Association Factors for’ to ‘Factors Associated with’ both in the tile and in the manuscript.

  1. Line 57: " We thought that the results... " should be changed to " We hoped that the results...". Or "We anticipated that the findings of this study would enhance management strategies for osteoarthritis (OA) and ultimately improve the health-related quality of life in older adults affected by OA."

 -> We appreciated your comment. We changed that sentence according to your comment. (line 56-58)

  1. Line 133: "SPSS ver. 21.0 for Windows" should be "IBM SPSS ver. 21.0 for Windows"

 -> We appreciated your comment. We changed that sentence according to your comment. (line 132)
